# Multi-Criterion Sampling Matting Algorithm via Gaussian Process

**DOI:** 10.3390/biomimetics8030301

**Published:** 2023-07-10

**Authors:** Yuan Yang, Hongshan Gou, Mian Tan, Fujian Feng, Yihui Liang, Yi Xiang, Lin Wang, Han Huang

**Affiliations:** 1Guizhou Key Laboratory of Pattern Recognition and Intelligent System, Guizhou Minzu University, Guiyang 550025, China; 18886302655@163.com (Y.Y.);; 2School of Computer Science, Zhongshan Institute, University of Electronic Science and Technology of China, Zhongshan 528400, China; 3Key Laboratory of Big Data and Intelligent Robot (SCUT), MOE of China, School of Software Engineering, South China University of Technology, Guangzhou 510006, China

**Keywords:** computing resources, Gaussian process fitting model, multi-criterion sampling strategy, high-quality pixel pairs, alpha matte

## Abstract

Natural image matting is an essential technique for image processing that enables various applications, such as image synthesis, video editing, and target tracking. However, the existing image matting methods may fail to produce satisfactory results when computing resources are limited. Sampling-based methods can reduce the dimensionality of the decision space and, therefore, reduce computational resources by employing different sampling strategies. While these approaches reduce computational consumption, they may miss an optimal pixel pair when the number of available high-quality pixel pairs is limited. To address this shortcoming, we propose a novel multi-criterion sampling strategy that avoids missing high-quality pixel pairs by incorporating multi-range pixel pair sampling and a high-quality sample selection method. This strategy is employed to develop a multi-criterion matting algorithm via Gaussian process, which searches for the optimal pixel pair by using the Gaussian process fitting model instead of solving the original pixel pair objective function. The experimental results demonstrate that our proposed algorithm outperformed other methods, even with 1% computing resources, and achieved alpha matte results comparable to those yielded by the state-of-the-art optimization algorithms.

## 1. Introduction

Image matting is a crucial image processing technique with extensive applications in image synthesis [1,2], video editing [3,4], live broadcasting [5,6], and film special effects [7,8]. In image matting, alpha mattes can accurately extract foreground objects and merge them with new backgrounds to render new scenes [9,10]. The concept of the alpha matte was first proposed by Thomas in 1984 [11], whereby an alpha matte estimation model was constructed by introducing the alpha channel. Mathematically, the color value $I_{p}$, including the RGB, of a pixel p can be linearly represented by the foreground color $F_{p}$ and the background color $B_{p}$ in the original image, as shown in Equation (1):(1)$$I_{p}=\alpha_{p}F_{p}+(1-\alpha_{p})B_{p}\;$$
where $\alpha_{p}$ is the alpha matte of the foreground object at pixel p, and $\alpha_{p}\in [0,1]$. Specifically, $\alpha_{p}$ takes 1 when p belongs to the foreground, it is assigned a value of 0 when p belongs to the background, and assigned a specific value in the range (0,1) when p is a semi-transparent pixel of which the color is a mixture of a foreground pixel color and a background pixel color. As both $F_{p}$ and $B_{p}$ are three-dimensional unknown vectors and $\alpha_{p}$ is an unknown scalar, Equation (1) is an ill-defined problem. In order to accurately determine the value of α, Rhemann et al. [12] introduced a trimap, which divides the image into three non-overlapping regions, denoted as the known foreground, known background, and unknown regions, to impose additional constraints on the image matting problem, as shown in Figure 1 (where F is the foreground region with α=1, B is the background region with α=0, and U is the unknown region with α in the range (0, 1)).

Pixel pair optimization-based matting methods are a class of competitive image matting approaches that offer significant advantages in terms of parallelization [13]. These methods are particularly effective in processing an error-marked trimap [14] or a foreground that is spatially disconnected [15,16,17]. Essentially, the natural image matting problem is transformed into a pixel pair optimization problem, as shown in Equation (2).
(2)$$\begin{array}{l} \min f(x_{p})\\ s.t.x_{i}={\left(x_{i}^{\left(F\right)},x_{i}^{\left(B\right)}\right)}^{T}\\ \;p\in U,\;x_{i}^{\left(F\right)}\in F,\;x_{i}^{\left(B\right)}\in B \end{array}$$
where $f(x_{p})$ is the pixel pair evaluation function of unknown pixel p; U,F, and B are the pixel sets of the unknown, known foreground, and known background regions, respectively; $x_{p}$ is the pixel pair decision vector of pixel p; $x_{i}^{F}$ and $x_{i}^{B}$ represent pixels in the known foreground and background regions, respectively. Once the foreground and background colors are obtained by solving the pixel pair optimization problem, the alpha value of the pixel p can be estimated via the following expression: (3)$${\hat{\alpha }}_{p}=\frac{\left(I_{p}-B_{p})(F_{p}-B_{p}\right)}{{\left\|F_{p}-B_{p}\right\|}^{2}}\;$$
where ${\left\|*\right\|}^{2}$ denotes the Euclidean norm of vector ∗.

Pixel pair optimization-based matting methods can be further divided into sampling-based methods and evolutionary optimization-based matting methods. Sampling-based methods [18,19] to evaluate the alpha value of the unknown region by collecting the known foreground and background pixels as candidate samples, which narrows the search range through different sampling strategies. For example, Liang et al. [20] proposed a surrogate model based on natural image matting that effectively reduces computational resource consumption by building on top of random sampling. However, this approach also faces the issue of the potential loss of optimal pixel pairs due to the random sampling strategy. The main drawback of these sampling-based methods stems from the significant likelihood of high-quality pixel pair loss, resulting in unsatisfactory alpha mattes. To avoid this issue, He et al. [21] proposed a global sampling method that employs all pixel pairs as candidate samples to avoid the loss of high-quality pixel pairs. However, this approach also results in increased computational resource consumption. Then, the evolutionary optimization-based [22,23] methods were proposed, which can effectively mitigate the drawback of high-quality pixel pair loss in sampling-based methods and improve the quality of alpha mattes. For instance, Liang et al. [24] used the evolutionary algorithm instead of the pixel pair sampling process and proposed a multi-objective evolutionary algorithm based on multi-criteria decomposition. This algorithm utilizes all available computing resources by adjusting the number of iterations of the evolutionary algorithm and theoretically eliminates the risk of missing real samples. On the other hand, to address the problem of computing resource consumption, Liang et al. [25] developed a multi-scale evolutionary pixel pair optimization framework, which transforms the large-scale pixel-pair optimization problem into multiple sub-optimization problems of different scales by using an image pyramid. Although evolutionary optimization-based methods utilizing various evolutionary algorithms have improved the accuracy of the alpha matte, it may require thousands of iterations to find the optimal solution. Limited computing resources may restrict the applicability of this method and also compromise the quality of alpha mattes. In summary, neither the sampling-based method nor the existing evolutionary optimization-based method can provide satisfactory alpha mattes under limited computing resources. Regarding image matting tasks, there has been less discussion on matting under limited computing resources. 

These limitations have motivated the present study, in which we designed a multi-criterion sampling strategy (MCSS) to ensure that high-quality pixel pairs are sampled. Furthermore, to reduce the consumption of computing resources, the multi-criterion matting algorithm via Gaussian process (GP-MCMatting) is proposed, which can provide a satisfactory alpha matte even when computing resources are limited. The contributions of the work presented in this paper are threefold: By combining different features in a multi-criteria sampling strategy (MCSS), the problem of missing high-quality pixel pairs is alleviated;This paper changes the traditional matting method, which only relies on one evaluation function, and combines multiple evaluation functions to comprehensively evaluate pixel pairs to select high-quality pixel pairs, avoiding the limitation of a single evaluation function;In order to ensure that the matting problem can be solved even with limited computing resources, a new perspective was adopted. This paper proposes a new GP-MCMatting algorithm, in which we use the Gaussian process fitting model (GPFM) instead of the objective function to search for the optimal pixel pair. By using this algorithm, effective and accurate matching can be achieved with only 1% of computing resources.

Next, a description of the algorithm proposed in this article is presented. The article consists of three sections: Section 2 covers related work, Section 3 outlines the problem, and Section 4 introduces the GP-MCMatting algorithm, which includes the MCSS in Section 4.1 and the Gaussian process fitting model in Section 4.2.1. The related work in this field is presented first.

## 2. Related Work

Recently, the field of image matting has experienced significant advancements, as reflected by numerous publications that can be broadly categorized into those related to sampling-based methods and those related to evolutionary optimization-based methods. In the brief literature review provided below, we focus on the work most closely related to this article. 

Sampling-based methods: Sampling-based methods take the sample pixel pairs of the known regions as the candidate sample and select the optimal foreground/background pixel pair through the evaluation function to solve the alpha matte. For example, to avoid the loss of real samples, Feng et al. [26] clustered foreground/background regions and selected representative pixels in each class as candidate samples. Inspired by image inpainting, Tang et al. [19] combined sampling with deep learning methods and used the image inpainting network to select foreground/background pixel pairs as candidate samples to evaluate the alpha mattes of unknown regions. As a part of their research, Huang et al. [27] designed a discrete multi-objective optimization algorithm based on pixel-level sampling. They, thus, effectively solved the problems of incomplete sampling space and optimal sample loss in the super-pixel sampling method and ensured the accuracy of the alpha matte. The strategy adopted by Cao et al. [28] was rooted in the patch-based image matting method, as these authors used a patch-based adaptive matting algorithm for high-resolution images and videos. As shown in their work, the algorithm extends the adaptive framework to video matting and reduces the consumption of computer resources. In short, in sampling-based methods, different sampling strategies [29,30,31] are utilized to select the sample pixel pairs, which reduces the scale of the decision space as well as the computational resource consumption. However, when there are few high-quality pixel pairs in the input image, the sampling-based methods may not identify the optimal pixel pair, resulting in an unsatisfactory alpha matte extraction accuracy. 

Evolutionary optimization-based methods: Most evolutionary optimization-based methods [32] are based on the assumption that adjacent pixels have similar alpha values. For example, Liang et al. [33] modeled the image matting problem as a combinatorial optimization problem in which foreground and background pixel pairs are assumed to be known. These authors designed a heuristic optimization algorithm based on an adaptive convergence speed controller, which alleviated the problem of premature algorithm convergence when solving for the optimal pixel pair. In their work, Feng et al. [34] focused on addressing the large-scale problem of high-definition images and proposed a competitive swarm optimization algorithm based on group collaboration to realize the group collaborative solution of large-scale combinatorial optimization problems. These authors demonstrated that, compared to the sampling-based method, the evolutionary optimization-based method could effectively improve the alpha matte estimation accuracy. However, when this method is adopted, more than $5\times 10^{3}$ pixel pairs need to be evaluated for each unknown pixel to ensure that high-quality pixel pairs are captured. Therefore, when the computing resources are limited, it is difficult to attain an adequate alpha value. In summary, when computing resources are limited, it is difficult to obtain a desired matting result with the existing evolutionary optimization-based and learning-based matting methods. Therefore, in this work, we propose a sampling-based multi-criterion sampling strategy (MCSS) to avoid the loss of high-quality pixel pairs. In addition, to achieve adequate matting with limited computing resources, we propose a novel algorithm—the multi-criterion matting algorithm via Gaussian process (GP-MCMatting)—incorporating the MCSS. The GP-MCMatting algorithm uses the Gaussian process fitting model (GPFM) instead of the original objective function to search for the optimal pixel pair, which effectively achieves matting under limited computing resources while ensuring the desired matting accuracy. 

## 3. Problem Description

The authors of the existing sampling-based matting methods have utilized different strategies to select samples from the known regions (the foreground region F and the background region B in Figure 1b) to obtain candidate samples for pixels in the unknown region (the unknown region U in Figure 1b), thereby evaluating the alpha matte of the unknown region. One such sampling method is shown in Figure 2. 

In the image above, I represents an RGB image, and F,B,U correspond to the known foreground, background, and unknown regions in the trimap respectively. S is the decision space composed of foreground and background samples, which is denoted by $S_{F}\times S_{B}$. Finally, the evaluation function is used to determine the fitness value of each pixel pair, and the pixel pair with the best fitness is selected as the optimal pixel pair $x_{opt}$, based on the criteria shown in Equation (4):(4)$$\begin{array}{l} \min f(x)=(x_{i}^{F},x_{i}^{B})\\ s.t.\;x\in U,(x_{i}^{F},x_{i}^{B})\in S\;\\ \;S=S_{F}\times S_{B} \end{array}$$
where f(∗) is the evaluation function of the foreground/background pixel pair, x denotes the unknown pixel in the unknown region, $\left(x_{•}^{F},x_{•}^{B}\right)$ represents the decision variable corresponding to the pixels in the known foreground and background regions, and $S_{F}\times S_{B}$ is the Cartesian product of the foreground and background sample sets. 

As the dimensions of the candidate sample obtained by sampling are smaller than those of the original decision space, sampling-based methods reduce the consumption of computing resources to a certain extent. However, when the number of high-quality pixel pairs in natural images is small (as shown in Figure 3), this strategy is prone to losing high-quality pixel pairs, resulting in an unsatisfactory matting quality.

As shown in Figure 3, depicting GT16 from the alpha matting dataset (with an 800 × 536 size), for the flag of the unknown region to be solved, there are fewer than 500 high-quality pixels confined to a smaller region. Thus, the probability of high-quality pixels being sampled is 0.0034. The sampling strategy proposed in this work that can be adopted to overcome this issue is described below.

## 4. Multi-Criterion Matting Algorithm via Gaussian Process 

In this section, we introduce the GP-MCMatting algorithm, which mainly consists of two stages: (1) the multi-criteria sampling strategy (MCSS); (2) the Gaussian process fitting model (GPFM).

### 4.1. Multi-Criterion Sampling Strategy

This subsection details the multi-criterion sampling strategy (MCSS), which consists of (1) multi-range pixel pair sampling and (2) high-quality sample selection.

#### 4.1.1. Multi-Range Pixel Pair Sampling

Sampling-based methods rely on local or global sampling strategies to sample pixel pairs, whereby the former approaches the risk of missing high-quality pixel pairs. In order to compensate for this shortcoming, when the latter strategy is adopted, all foreground and background pixel pairs are treated as the sample set to ensure that the optimal solution is found. While this avoids the loss of real samples, it also increases the consumption of computing resources. In this work, we mitigate the aforementioned shortcomings by assuming that there is an optimal pixel pair region in the input image, due to which high-quality pixel pairs can be identified by sampling this region. Guided by this assumption, we designed a multi-range pixel pair sampling method based on the color and spatial position of pixels. In multi-range pixel pair sampling, the color similarity score between the pixels can be reformulated as follows:(5)$$D^{c}={\left\|I_{p}-I_{i}^{U}\right\|}_{2},\;I_{p}\in F\cup B$$
where ${\left\|*\right\|}_{2}$ denotes the Euclidean norm of vector ∗, F∪B denotes the set of known regions, $I_{p}$ is the color value of the known pixel p, and $I_{i}^{U}$ is the color value of the unknown pixel i.

In image matting, the spatial positions of pixels can reflect their structural features, which may help in the differentiation of the pixels. For this purpose, the similarity between the unknown region and the known boundary pixels, as well as the similarity between the unknown region and the long-range known regions (as shown in Figure 4), are calculated using the expressions shown below:(6)$$D^{s}={\left\|I_{p,in}-I_{{}_{i,in}}^{U}\right\|}_{2},\;I_{p,in}\in F_{in}\cup B_{in}\;$$
(7)$$D^{b}={\left\|I_{p,in}^{b}-I_{{}_{i,in}}^{U}\right\|}_{2},\;I_{p,in}^{b}\in F_{in}\cup B_{in}\;$$
where $D^{s}$ denotes the spatial position distance between the unknown pixels and the long-range known pixels; $D^{b}$ represents the spatial distance between the pixels in the unknown region and the pixels in the known boundaries; $I_{p,in}^{b}$ represents the coordinate index values of the pixels in the known boundaries, $I_{p,in}$ denotes the coordinate index value of pixels in the long-range known region; $I_{i,in}^{U}$ denotes the coordinate index value of the unknown pixel, and $F_{in}\cup B_{in}$ is the coordinate space of the known regions.

Let the color distance set between the unknown pixels and the known pixels be given by $D^{c}=\{D_{1}^{c},D_{2}^{c},\ldots ,D_{m}^{c}\}$, whereas the spatial distance between the unknown pixels and the long-range known region pixels is given by $D^{s}=\{D_{1}^{s},D_{2}^{s},\ldots ,D_{m}^{s}\}$, and the spatial distance between the unknown pixels and the pixels in the known boundaries is $D^{b}=\{D_{1}^{b},D_{2}^{b},\ldots ,D_{m}^{b}\}$. For a more intuitive view of the pixel distance scores, these three sets are rearranged in ascending order, i.e., $D_{{}^{\left(\cdot \right)}}^{c}=\{D_{\left(1\right)}^{c},D_{\left(2\right)}^{c},\ldots ,D_{\left(m\right)}^{c}\}$, $D_{\left(\cdot \right)}^{s}=\{D_{\left(1\right)}^{s},D_{\left(2\right)}^{s},\ldots ,D_{\left(m\right)}^{s}\}$, and $D_{\left(\cdot \right)}^{b}=\{D_{\left(1\right)}^{b},D_{\left(2\right)}^{b},\ldots ,D_{\left(m\right)}^{b}\}$. According to the sample sets of three different distances, the color is first used as the main collection feature, after which the pixels with similar colors and close distances in the set are considered as the first type of candidate samples. Then, the sampling process is described by the following expressions:(8)$$X^{c}=\left\{ \begin{array}{l} X^{c}\cup S_{k}^{c},\;S_{k}^{c}<\frac{d*\max(S^{c})}{\left|S^{c}\right|}\\ \emptyset \;,\;other \end{array}\right.,\;k=1,2,\ldots ,\leq \left|S^{c}\right|\;$$
(9)$$S^{c}=\left\{ \begin{array}{l} S^{c}\cup \{D_{\left(j\right)}^{s}\}\cup \{D_{\left(j\right)}^{b}\},\;if\;D_{\left(j\right)}^{c}<\varepsilon \\ \emptyset \;,\;other \end{array}\right.,\;j=1,2,\ldots ,\left|D_{\left(\cdot \right)}^{c}\right|\;$$
where $X^{c}$ is a sample set with similar colors and close distances, whereas $S^{c}$ is a set of pixels with a color similar to the unknown pixels, ε denotes the threshold of the color distance, and $\frac{d*\max(S^{c})}{\left|S^{c}\right|}$ represents the proportion of pixels with a small color distance and coordinate space distance in the $S^{c}$ set, where d denotes the number of samples. 

Furthermore, according to the obtained multi-range sets, the coordinate space is the main acquisition feature, and the pixels in the same set characterized by a close spatial distance and similar color are considered as the second candidate sample type. This sampling method is described by the following equations:(10)$$X^{s}=\left\{ \begin{array}{l} X^{s}\cup S_{k}^{s},\;S_{k}^{s}<\frac{d*\max(S^{s})}{\left|S^{s}\right|}\\ \emptyset \;,\;other \end{array}\right.,\;k=1,2,\ldots ,\leq \left|S^{s}\right|$$
(11)$$S^{s}=\left\{ \begin{array}{l} S^{s}\cup \{D_{\left(j\right)}^{c}\},\;if\;D_{\left(j\right)}^{s}<d\\ \emptyset \;,\;other \end{array}\right.,\;j=1,2,\ldots ,\left|D_{\left(\cdot \right)}^{c}\right|$$
where $X^{s}$ is a sample set with a close spatial distance and similar colors, and $S^{s}$ represents a set of similar pixel distances. Thus, when sampling based on different primary features, the sample sets $X^{c}$ and $X^{s}$ are formed, allowing the final candidate sample X to be obtained as their union, i.e., $X=X^{c}\times X^{s}$ which is the Cartesian product of $X^{c}$ and $X^{s}$.

#### 4.1.2. High-Quality Sample Selection

As explained in the preceding section, multi-range pixel pair sampling yields the candidate sample set $X=X^{c}\times X^{s}$. In cases with a large number of candidate samples that need to be considered for obtaining high-quality pixel pairs, the high-quality sample selection method proposed here can be adopted, which is combined with the multi-range pixel pair sampling method to form the MCSS. In the process of pixel pair evaluation, a single evaluation function is usually used to evaluate the optimal pixel pair. However, this may compromise the ability to determine the fitness value of a given pixel pair [34]. Therefore, we propose a novel approach that combines multiple evaluation functions to select high-quality pixel pairs as candidate samples. Specifically, we employ color difference evaluation $f_{c}(x)$, fuzzy evaluation $f_{f}(x)$, and fuzzy multi-criterion evaluation $f_{m}(x)$ as joint evaluation indices for sample assessment. Let $f_{i}$ denote the fitness value after averaging the three evaluation functions of pixel x. Accordingly, the fitness value corresponding to the elements in candidate sample X is $F=\{f_{1},f_{2},\ldots ,f_{\left|X\right|}\}$, where $f_{i}=1/3\sum \left(f_{c}(x)+f_{f}(x)+f_{m}(x)\right)$. Reordering F in ascending order yields $F_{\left(\cdot \right)}=\{f_{\left(1\right)},f_{\left(2\right)},\ldots ,f_{\left(\left|X\right|\right)}\}$. Then, according to the fitness value obtained by the joint evaluation function, the high-quality pixel pairs can be selected using the expression below:(12)$$\Omega_{f}=\left\{ \begin{array}{l} \Omega_{f}\cup f_{i},\;if\;\min f_{i}(x)+1>f_{i}(x),\;i=1,2,\ldots ,\left|F\right|,\;f_{i}\in F\\ \emptyset \;,\;other \end{array}\right.\;\Omega_{f}\to \Omega$$
where $\Omega_{f}$ represents the set of fitness values of the high-quality pixel pairs; Ω is the pixel information corresponding to the fitness values of high-quality pixel pairs; and threshold $\min(f_{i}(x))+1$ is implemented to exclude excessively high values of the evaluation function. After obtaining high-quality pixel pairs according to Equations (12) and (13), the number of high-quality pixel pair samples is far smaller than the number of pixels in the decision space. As shown in Algorithm 1, the aforementioned approaches are combined to yield the MCSS.
**Algorithm 1:** Multi-criterion sampling strategy**Input:** image and trimap.1.//For information on image pixel.2. F∪B= Known pixel information by trimap.3.
**for** i=1
 **to** 
|F∪B|
 **do**
4.      //multi-range pixel-pairs sampling.5.      $\{D^{c},D^{s},D^{b}\}=$ By Equations (5)–(7).6.      $\{D_{\left(\cdot \right)}^{c},D_{\left(\cdot \right)}^{s},D_{\left(\cdot \right)}^{b}\}=$ Sort the sets in ascending order.7.      **while** $D_{i,(\cdot )}^{c}<\varepsilon$  **do**8.                $X^{c}=$ By Equations (8) and (9).
9.      **end while**

10.    **while**
 $D_{\left(\cdot \right)}^{s}<d$ 
**do**
11.              $X^{s}=$ By Equations (10) and (11).
12.     **end while**

13.    $X=X^{c}\times X^{s}$

14.**end for**
15.//High-quality sample selection.16.$F_{\left(\cdot \right)}=\{f_{\left(1\right)},f_{\left(2\right)},\ldots ,f_{\left(\left|X\right|\right)}\}=$ By Equation (12).17.**if** stop criterion is not met **then**18.        $\Omega =\Omega \cup \{x_{i}\}$  $//x_{i}$ $\mathrm{is}\;\mathrm{the}\;\mathrm{corresponding}\;\mathrm{sample}\;\mathrm{pixel}\;\mathrm{of}\;f_{i}$.
19. **end if**
**Output**: high-quality pixel pairs set  Ω.

Specifically, to ensure that high-quality pixel pairs are not missing, in Lines 1–2, the foreground and background pixel information is gathered according to the input image, while in Lines 3–14, the candidate sample set X is output after excessive range sampling, and in Lines 15–19, the final high-quality sample set Ω is obtained according to the high-quality selection method.

### 4.2. Multi-Criterion Matting Algorithm via Gaussian Process

In this subsection, we introduce the multi-criterion matting algorithm via Gaussian process (GP-MCMatting) proposed in this work, which enables image matting even when computing resources are limited while ensuring the required accuracy of the image matting.

#### 4.2.1. Gaussian Process Fitting Model

The objective function of the natural image matting problem is typically complex due to the simultaneous consideration of the similarity between pixels in the foreground and background regions, as well as the similarity among pixels. Traditional objective functions usually require nonlinear optimization algorithms for solving, which demands intensive computational resources and time [25]. When computing resources are limited, it may be challenging to directly use traditional objective function-based methods. Accordingly, we assume that approximating the evaluation of the objective function by fitting the parameters of Gaussian processes (GPs) [20] can improve the accuracy of alpha matte. Therefore, the GPFM can be represented as follows:(13)$$\begin{array}{l} \min H(x_{i,in})\to \min f(x)\\ s.t.\min f(x)\\ \;x=(x_{i}^{F},x_{i}^{B}),\;x_{i,in}=(x_{i,in}^{F},x_{i,in}^{B})\\ \;x_{i}^{F},x_{i,in}^{F}\in F;\;x_{i}^{B},x_{i,in}^{B}\in B \end{array}$$
where $H(x_{i,in})$ is the Gaussian process fitting model (GPFM). Because there are many redundant solution space regions in the coordinate space, in the proposed algorithm, the image index number space is used to construct the distribution of the objective function. Moreover, to avoid crossing the boundary when searching for the optimal solution, the GPFM is constructed in the index value space. 

The GP is an infinite set of random variable distributions, where the joint distribution function of each finite subset is subject to Gaussian distribution [35,36]. Therefore, the GP is completely determined by the mean function and the covariance kernel function between any two random variables [37,38]. In this work, the optimization of the GPFM model is achieved by approximating the optimal solution of the pixel pair objective function, essentially by estimating the undetermined parameters in the GPFM and then using the evaluated parameters to obtain the optimal solution [20]. Let θ and $\hat{\theta }$ denote the parameter vector and the estimated parameter vector of the kernel function in the GPFM, respectively. Then, the likelihood function is used to estimate the kernel function parameters in the Gaussian process fitting model. Mathematically, the expression can be described as follows:(14)$$\hat{\theta }=L(\theta |f(x),x_{i,in})=\underset{\theta }{\arg\min}\log P(f(x),x_{i,in}|\theta )$$

Therefore, according to Equation (15), the GPFM of the pixel pair evaluation function can be obtained via the following expressions:(15)$$\min H=H(x_{i,in}|\hat{\theta })$$

Then, the index position of the optimal pixel pair can be evaluated by fitting the GPFM in Equation (16), allowing us to solve for the optimal pixel pair. 

#### 4.2.2. Multi-Criterion Matting Algorithm via Gaussian Process

Natural image matting is a large-scale problem, for which extensive computing resources are needed to solve the original objective function directly. Therefore, to achieve matting with limited computing resources, we propose a novel matting algorithm named the multi-criterion matting algorithm via Gaussian process (GP-MCMatting), which includes (1) the multi-criterion sampling strategy (MCSS) and (2) the Gaussian process fitting model (GPFM). The former is mainly used to select high-quality sample pixel pairs from a large number of decision variables. The latter mainly uses the GPFM to approximate the original objective function of natural images based on the MCSS, thereby reducing the consumption of computational resources. For constructing the GPFM basis, we define Θ as a penalty function of the GPFM: (16)$$\Theta (\Omega ,\gamma^{\left(k\right)})=H(\Omega )-\gamma^{\left(k\right)}\sum \log f(\Omega )$$
where $\gamma^{\left(k\right)}$ is the penalty factors that decrease sequentially and $\underset{k\to +\infty }{\lim}\gamma^{\left(k\right)}=0$, Ω is the set of high-quality pixel pair samples. 

Once the GPFM is constructed, it is utilized to approximate the original objective function for a given pixel pair, allowing the optimal pixel pair to be obtained by solving the extreme points of the model via the high-quality pixel pair set Ω. In this work, the optimal solution of GPFM is obtained based on the ideas behind the interior-point algorithm (IPA) [39]. Within the solving process, the optimal solution $x_{\min}$ of the GPFM is approximately equivalent to the optimal solution $x_{best}$ of the objective function. In addition, we find that the optimal pixel pair is usually located in the $X^{loc}$ of the known region closest to the unknown pixels. Therefore, selecting the closest pixel pair in the local foreground region and the background region as the initial IPA population often yields a viable solution. When this assumption does not hold, the IPA algorithm can also find the optimal foreground/background pixel pairs by solving the GPFM, because the algorithm is applied to the entire search space, thus theoretically avoiding the loss of the optimal pixel pair. The solving process of GP-MCMatting is shown in Algorithm 2. 

Specifically, let ϵ represent the distance between the GPFM and the objective function that needs to be approximated, and τ is smaller than a specified small value. $\frac{\partial \Theta }{\partial x^{\gamma^{\left(k\right)}}}$ is the partial derivative, where c is a randomly generated attenuation factor, and $\gamma^{\left(k\right)}$ is a penalty coefficient greater than 0. Ω is a set of high-quality pixel pairs obtained by Algorithm 1. In Algorithm 2, the required parameters are first initialized, and Lines 5–10 are used to construct GPFM. Lines 12–22 are the process of solving the optimal solution of the GPFM. The algorithm stops iterating when the optimal solution of the GPFM is approximately equivalent to the optimal solution of the objective function.
**Algorithm 2:** Multi-criteria matting algorithm via Gaussian process**.****Input:** image and trimap.1. //Initialize parameters ϵ,τ.2.
$x^{loc}=X^{loc}(x).$
3.$\gamma^{\left(k\right)}=$ a random number greater than 0.
4.
c= a random number between [0, 1].
5.**for** 
i=1
 **to** 
|U|
 **do**
6.      //Gaussian process fitting model construction.7.      Ω= According to the Algorithm 1.8.      $(f(x),x_{i,in})=\Omega$.
9.      $H=H(L_{p}(\theta |f(x),x_{i,in})),x_{i,in}\in \Omega$
**.**
10.    //Optimal pixel pair estimation.
11.    $x_{\min}=\min H(x^{loc})$
**.**

12.    **while **
ϵ>τ
 **do**
 13.              xγ(k)=∂Θ∂xγ(k)=0.14.xinit=xγ(k).15.ϵ=‖xγ(k)−xinit‖.16.γ(k)=γ(k−1)⋅c.17.               **if**
γ(k)≤0.  **then**18.                      γ(k)= a random number greater than 0.
19.c= a random number between [0, 1]. 20.**end if**
21.    **end while**

22.**end for**23.
$x_{best}=x^{\gamma^{\left(k\right)}}$
**Output**$:\;x_{best}$.

## 5. Experiments and Results

In this section, three sets of experiments are described, which were performed to verify the effectiveness of the high-quality pixel pair selection strategy and the effectiveness of the algorithm.

### 5.1. Experimental Setup

In the experiments, the alpha matting dataset presented by Rhemann et al. [9] (which contains 35 images, 27 of which are training images and 8 are testing images) was used as the benchmark dataset. All experiments were implemented in MATLAB, and we set up three groups of experiments, as outlined below. 

This experiment was mainly performed to verify that the MCSS can effectively avoid the loss of high-quality pixel pairs, thus improving the matting performance. In this experiment, a comparative analysis was conducted between GP-MCMatting and the state-of-the-art evolutionary optimization-based algorithms, such as the pyramid matting framework (PMF) [25], adaptive convergence speed controller based on particle swarm optimization (PSOACSC) [33], and the multi-objective evolutionary algorithm based on multi-criteria decomposition (MOEAMCD) [24], based on 1%, 2%, 5%, 10%, 20%, and 100% computing resources. The matting performance of the proposed algorithm under limited computing resources was verified. This experiment was conducted to compare the GP-MCMatting performance with the aforementioned algorithms based on the availability of only 1% computing resources to verify its superiority.

In order to ensure the comparability of the experimental results based on different algorithms and to facilitate statistical analysis of their performance, the mean square error (MSE) and absolute value error (SAD) were adopted as the evaluation indices. In the sections that follow, the results and analyses of each group of experiments are presented.

### 5.2. Effectiveness of Multi-Criterion Sampling Strategy 

In the MCSS, the spatial coordinates and color features of the pixels were sampled by solving Equations (8) and (9). Using Equation (9), pixels whose color distance was less than ε were considered candidate samples. In order to determine the optimal value of ε, we calculated the matting results under different distances. Figure 5 shows the sum of the MSE values of the 27 alpha matting training sets based on 1% computing resources and the different values of ε. The results show that when the color distance parameter ε was assigned the value of 10 (the red corresponds to the dashed line), the total MSE based on all 27 images was the smallest with a value of 0.789, i.e., the alpha matte accuracy was the highest. Therefore, the color distance parameter ε was set to 10 in the MCSS.

Additionally, to verify the effectiveness of the high-quality sample selection method in the MCSS, a comparative analysis of the matting quality obtained by selecting different evaluation functions was performed, and the findings are shown in Table 1. The comparisons were based on the sum of the MSE and SAD of the 27 images solved by different evaluation functions. By referring to Table 1, we can see that on the basis of the MSE and SAD evaluation metrics, the MCSS outperformed the method based on a single evaluation function. Therefore, the experimental results show that the combination of multiple evaluation functions to select high-quality pixel pairs was significantly better than a single evaluation function to select pixel pairs, which fully verifies the effectiveness of the MCSS.

### 5.3. Algorithm Evaluation and Comparison under the Conditions Characterized by Limited Computing Resources 

In this subsection, GP-MCMatting is compared with the state-of-the-art evolutionary optimization-based algorithms to verify the matting performance of the proposed algorithm when computing resources are limited. As most algorithms require 5000 iterations per pixel [24,25,33], this was adopted as the benchmark, and 1%, 2%, 5%, 10%, 20%, and 100% availability of computing resources were considered in 5000 different scenarios to compare the matting performance of the studied algorithms. For the fairness of comparison, the MSE and SAD were used as the evaluation indicators to compare the alpha matte accuracy of the different algorithms. The results in Table 2 represent the sum of the MSE and SAD of different algorithms when applied to the training sets contained in the alpha matting dataset under different computing-resource conditions.

According to the results reported in Table 2, in terms of the SAD value, the matting performance of the proposed algorithm under limited and sufficient computing resources was superior to those associated with the PMF, MOEAMCD, and PSOACSC algorithms. According to the MSE evaluation index, the matting effect of the proposed GP-MCMatting algorithm was better than that of the algorithms used in the comparison only when 1%, 2%, and 5% of the computing resources were considered. Nonetheless, the GP-MCMatting algorithm was still superior to the PMF and PSOACSC in the scenarios based on 10%, 20%, and 100% computing resource availability. When both the SAD and MSE were considered, however, the proposed GP-MCMatting algorithm not only achieved competitive matting effects under limited computing resources, but also achieved good results even when computing resources were sufficient. Thus, the proposed algorithm based on the MCSS effectively realized the matting results irrespective of the computing resource availability. 

In order to further analyze the effectiveness of GP-MCMatting under limited computing resources, the MSE results of different algorithms were compared based on their application on 27 images in the alpha matting dataset and the availability of 1% computing resources.

### 5.4. Comparison to the State-of-the-Art Methods

In order to demonstrate the effectiveness of the proposed GP-MCMatting algorithm and compare its performance to the state-of-the-art algorithms when computing resources are limited, the MSE values obtained by applying different matting algorithms to the 27 training sets under 1% computing resources are shown in Table 3. 

According to the results reported in Table 3, GP-MCMatting outperformed the PMF, PAOACSC, and MOEAMCD algorithms in 24 of the 27 cases (with the exception of the GT24, and GT15 images). The reasons behind its suboptimal performance in these two cases are analyzed in detail in Section 5.5.

To further verify the matting effect of the proposed algorithm, the results yielded by the GP-MCMatting algorithm on the training set and test set are depicted in Figure 6. It is evident that GP-MCMatting effectively evaluated the object containing hair in the foreground target in the image (the visualization results in rows 1, 2, 3, 4, and 6). Conversely, according to the comprehensive visualization results and the MSE accuracy results, the matting effect of the compared algorithms under the limited computing resources was not satisfactory, and there were large unknown areas that could not be evaluated. Based on the analysis of the experimental results shown in Table 3 and the images included in Figure 6, it is evident that the algorithm proposed in this work avoided the loss of the optimal pixel pair by selecting high-quality pixel pairs, thus improving the extraction accuracy of the alpha matte. In addition, under limited computing resources, satisfactory matting was realized by approximately solving the GPFM, which reflects the superiority of the proposed GP-MCMatting.

### 5.5. Limitations of the GP-MCMatting Algorithm

The comparison and analysis of the experimental results in Section 5.4 fully reflect the superiority of the proposed algorithm under conditions characterized by limited computing resources. However, the results reported in Table 3 in Section 5.4 show that the extraction accuracy of the GP-MCMatting algorithm was low when applied to the GT15 and GT24 images, and there were failures in the acquisition of high-quality pixel pairs, as shown in Figure 7. 

The results depicted in Figure 7 reveal that the color information of the foreground and background pixels in the annotated region was relatively similar, as depicted by the green box in the images. Although the pixels in this region met the color sampling requirements due to their minor color differences, the spatially close distribution of these pixels impeded the search for the optimal pixel pair during the evaluation of the kernel function parameters of the Gaussian process field model (GPFM) using pixel spatial distribution. Consequently, in complex images that exhibit similar foreground and background characteristics, GPFM approximations cannot replace the original evaluation function when solving for the optimal pixel pairs. This fundamental limitation constrains the practical implementation of the GP-MCMatting algorithm in such scenarios.

## 6. Conclusions

In this paper, we present a multi-criterion sampling matting algorithm via Gaussian process (GP-MCMatting) that effectively solves the matting problem with limited computing resources. To overcome the challenge of losing the optimal pixel pair, which occurs in sampling-based methods, an MCSS was designed based on multi-range pixel pair sampling and high-quality sample selection. Additionally, to address the challenge of matting under limited computing resources, the GP-MCMatting algorithm searches for the optimal pixel pair using an approximation of the GPFM instead of the original evaluation function. This technique avoids the protracted calculations required by the original evaluation function while still ensuring accurate matting. The experimental results demonstrate that the GP-MCMatting algorithm achieved similar results under sufficient computing resources and was superior when 1%, 2%, and 5% of computing resources were available.

However, when images with similar foreground and background color information were input, the GP-MCMatting algorithm failed to achieve satisfactory results. Therefore, to address the inherent limitations of the algorithm, future research will focus on exploring the color variation characteristics of adjacent pixels in images, taking into account the spatial continuity and correlation between pixels. This approach aims to solve the challenge of accurately extracting the alpha matte in images with extremely similar foreground and background color information, where the alpha matte is difficult to extract.

## Figures and Tables

**Figure 1 biomimetics-08-00301-f001:**
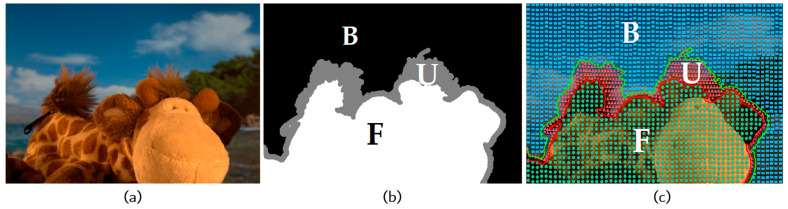
Image and trimap, where (**a**) Input image; (**b**) Trimap; (**c**) Constraint regions by trimap.

**Figure 2 biomimetics-08-00301-f002:**
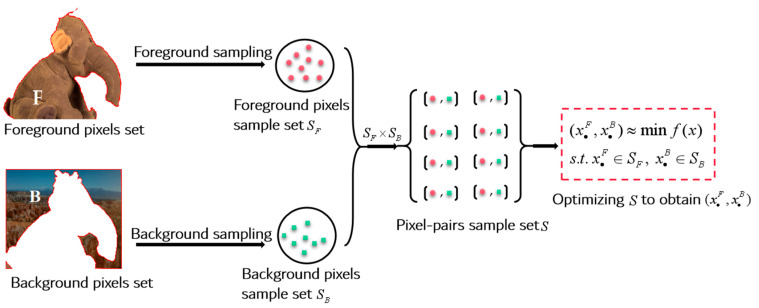
An example of a sampling-based matting method.

**Figure 3 biomimetics-08-00301-f003:**
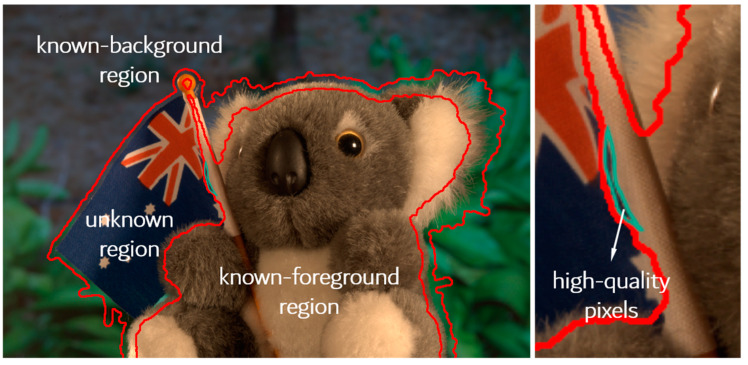
Case with a small number of high-quality pixels in the original image.

**Figure 4 biomimetics-08-00301-f004:**
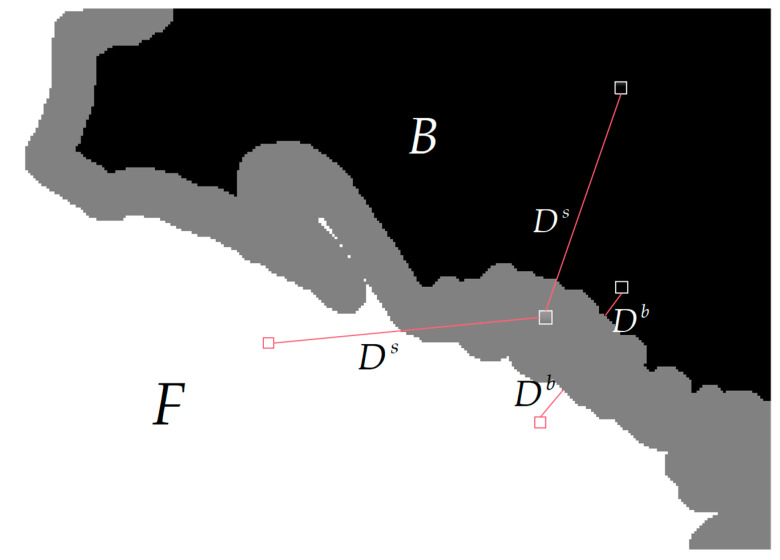
Spatial distance between pixels.

**Figure 5 biomimetics-08-00301-f005:**
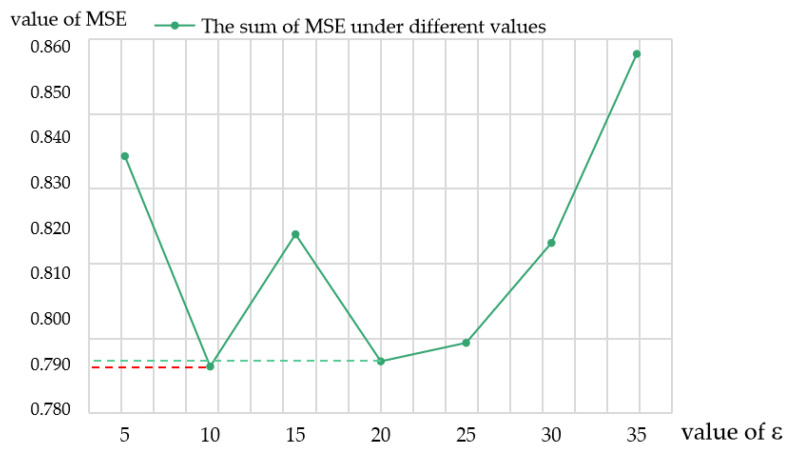
Sensitivity analysis of the color distance parameter.

**Figure 6 biomimetics-08-00301-f006:**
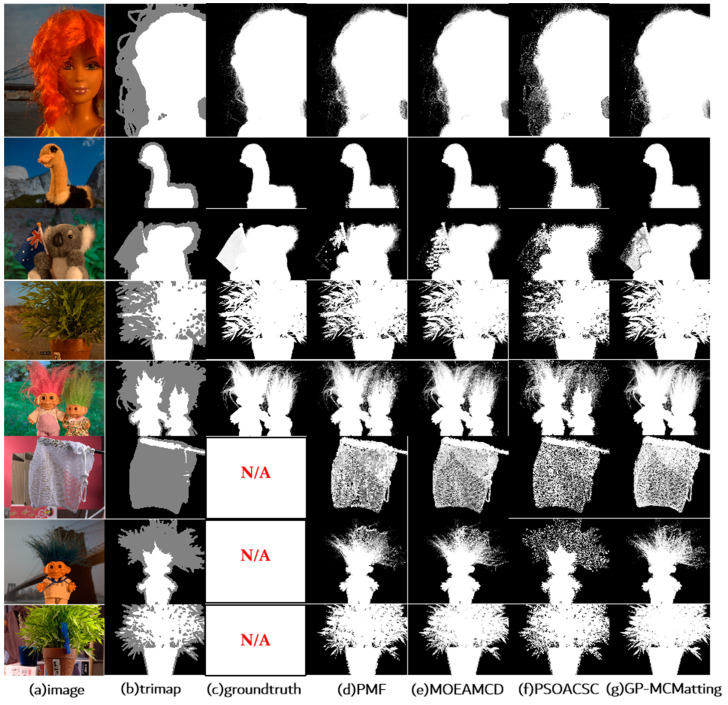
Visual comparison on the alpha matting dataset. (**a**) input image; (**b**) trimap; (**c**) groundtruth; (**d**) PMF; (**e**) MOEA-MCD; (**f**) PSOACSC; (**g**) GP-MCMatting.

**Figure 7 biomimetics-08-00301-f007:**
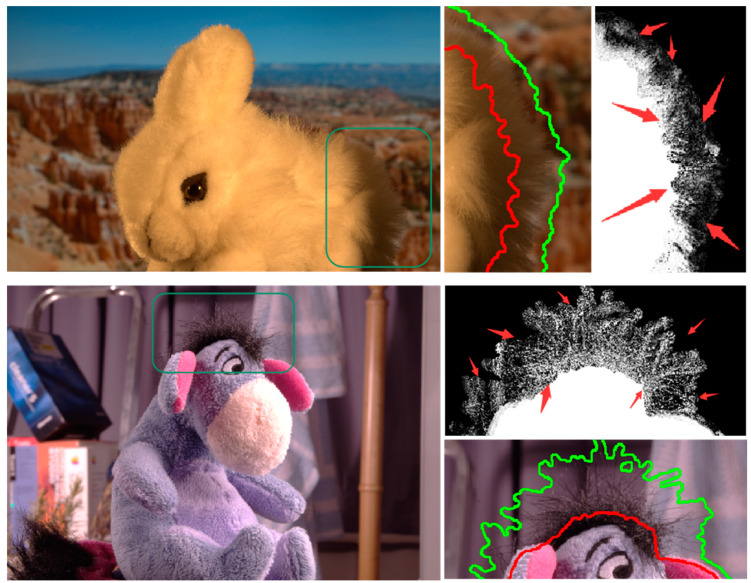
The case of GP-MCMatting failure when applied to the GT15 and GT24 images.

**Table 1 biomimetics-08-00301-t001:** Comparison of the matting results yielded by different evaluation functions.

	Color Difference Evaluation Function	Multi-Objective Evaluation Function	Fuzzy Evaluation Function	Multi-Criteria Sampling Strategy
MSE	0.0299	0.0297	0.0293	**0.0292**
SAD	7.2699	7.1731	7.1628	**7.0999**

**Table 2 biomimetics-08-00301-t002:** The SAD and MSE values for the GP-MCMatting and the state-of-the-art evolutionary optimization-based algorithms based on their application to 27 images at 1%, 2%, 10%, 20%, and 100% computer power and 5000 evaluations. Bold Arabic numerals indicate the index with the highest ranking.

Computing Resources	SAD					
	1%	2%	5%	10%	20%	100%
PSOACSC [33]	604.425	604.267	604.418	604.137	603.950	602.987
MOEAMCD [24]	243.965	243.346	242.741	242.179	243.028	242.741
PMF [25]	349.160	321.151	294.067	272.818	253.481	228.433
Ours	**191.697**	**187.752**	**202.500**	**214.409**	**216.114**	**210.910**
Computing resources	MSE					
	1%	2%	5%	10%	20%	100%
PSOACSC [33]	5.061	5.063	5.063	5.062	5.056	5.039
MOEAMCD [24]	1.127	1.121	1.113	**1.116**	**1.118**	1.113
PMF [25]	2.231	1.910	1.689	1.430	1.257	**1.028**
Ours	**0.789**	**0.826**	**1.029**	1.151	1.163	1.171

**Table 3 biomimetics-08-00301-t003:** The MSE values of GP-MCMatting and the state-of-the-art evolutionary optimization-based methods based on their application on 27 images under 1% computing resources. Bold Arabic numerals indicate the index with the highest ranking.

Algorithms	GT01	GT02	GT03	GT04	GT05	GT06	GT07	GT08	GT09
Ours	**3.33 × 10^−3^**	**6.40** ** × ** **10^−3^**	**9.68** ** × ** **10^−3^**	**1.23** ** × ** **10^−2^**	**1.49** ** × ** **10^−2^**	**1.49** ** × ** **10^−2^**	**7.89** ** × ** **10^−3^**	**3.93** ** × ** **10^−2^**	1.29 × 10^−2^
PSOACSC [33]	6.65 × 10^−2^	2.13 × 10^−1^	5.35 × 10^−2^	1.01 × 10^−1^	1.62 × 10^−1^	2.24 × 10^−1^	9.11 × 10^−2^	1.26 × 10^−1^	1.57 × 10^−1^
MOEAMCD [24]	7.49 × 10^−3^	1.53 × 10^−2^	1.19 × 10^−2^	2.15 × 10^−2^	2.53 × 10^−2^	2.71 × 10^−2^	1.08 × 10^−2^	4.40 × 10^−2^	**1.11** ** × ** **10^−2^**
PMF [25]	1.74 × 10^−2^	1.23 × 10^−1^	1.45 × 10^−2^	3.28 × 10^−2^	4.78 × 10^−2^	4.45 × 10^−2^	2.84 × 10^−2^	5.71 × 10^−2^	2.16 × 10^−2^
**Algorithms**	**GT10**	**GT11**	**GT12**	**GT13**	**GT14**	**GT15**	**GT16**	**GT17**	**GT18**
Ours	**2.32** ** × ** **10^−2^**	**3.21** ** × ** **10^−2^**	**1.00** ** × ** **10^−2^**	**1.23** ** × ** **10^−2^**	**7.81** ** × ** **10^−3^**	5.04 × 10^−2^	**7.64** ** × ** **10^−2^**	**1.22** ** × ** **10^−2^**	**7.50** ** × ** **10^−3^**
PSOACSC [33]	2.22 × 10^−1^	3.02 × 10^−1^	4.61 × 10^−2^	2.08 × 10^−1^	1.55 × 10^−1^	1.79 × 10^−1^	3.62 × 10^−1^	1.41 × 10−1	2.04 × 10^−1^
MOEAMCD [24]	2.77 × 10^−2^	3.93 × 10^−2^	1.47 × 10^−2^	2.29 × 10^−2^	2.61 × 10^−2^	**4.02** ** × ** **10^−2^**	1.87 × 10^−1^	1.59 × 10^−2^	1.66 × 10^−2^
PMF [25]	6.36 × 10^−2^	7.65 × 10^−2^	1.84 × 10^−2^	6.66 × 10^−2^	5.37 × 10^−2^	9.36 × 10^−2^	3.43 × 10^−1^	3.01 × 10^−2^	6.15 × 10^−2^
**Algorithms**	**GT19**	**GT20**	**GT21**	**GT22**	**GT23**	**GT24**	**GT25**	**GT26**	**GT27**
Ours	**8.82** ** × ** **10^−3^**	**7.56** ** × ** **10^−3^**	**1.57** ** × ** **10^−2^**	**7.15** ** × ** **10^−3^**	**7.35** ** × ** **10^−3^**	9.84 × 10^−2^	**1.51** ** × ** **10^−1^**	**5.95** ** × ** **10^−2^**	**8.06** ** × ** **10^−2^**
PSOACSC [33]	2.27 × 10^−1^	7.33 × 10^−2^	1.90 × 10^−1^	1.19 × 10^−1^	1.58 × 10^−1^	3.54 × 10^−1^	2.99 × 10^−1^	2.56 × 10^−1^	3.71 × 10^−1^
MOEAMCD [24]	5.13 × 10^−2^	1.21 × 10^−2^	4.73 × 10^−2^	1.38 × 10^−2^	1.81 × 10^−2^	**6.88** ** × ** **10^−2^**	1.67 × 10^−1^	6.94 × 10^−2^	1.15 × 10^−1^
PMF [25]	9.99 × 10^−2^	1.34 × 10^−2^	9.56 × 10^−2^	2.61 × 10^−2^	2.32 × 10^−2^	1.05 × 10^−1^	2.88 × 10^−1^	1.64 × 10^−1^	2.22 × 10^−1^

## Data Availability

The figures utilized to support the findings of this study are included in the article.

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
