# Peer review of "Multi-Criterion Sampling Matting Algorithm via Gaussian Process"

_biomimetics, 2023, doi:10.3390/biomimetics8030301_

Round 1
Reviewer 1 Report
The manuscript proposes a new multi criterion sampling strategy that combines multi range pixel pairing and high-quality sample selection methods to avoid losing high-quality pixel pairs. Using this strategy, a multi criterion masking algorithm based on Gaussian process is proposed, which uses Gaussian process to fit the model and search for the best pixel pair per unit, rather than solving the original pixel pair objective function.
The experimental results show that under 1% computational resources, the performance of the algorithm proposed in the manuscript even surpasses other methods, and achieves alpha masking results comparable to state-of-the-art optimization algorithms.
1. Regarding the issue of image annotations for the manuscript. In order to help readers better distinguish between the main text and annotations, it is recommended that the font of the annotations in the manuscript should be different from that in the main text, which can serve as a reminder to readers and make the boundaries of the article clearer.Please refer the recent papers: Chenggang Yan, Biao Gong, Yuxuan Wei, Yue Gao, “Deep Multi-View Enhancement Hashing for Image Retrieval”, IEEE Transactions on Pattern Analysis and Machine Intelligence, 2020.Chenggang Yan, Zhisheng Li, Yongbing Zhang, Yutao Liu, Xiangyang Ji, Yongdong Zhang, “Depth image denoising using nuclear norm and learning graph model”, ACM Transactions on Multimedia Computing Communications and Applications 2020.Chenggang Yan, Yiming Hao, Liang Li, Jian Yin, Anan Liu, Zhendong Mao, Zhenyu Chen, Xingyu Gao, “Task-Adaptive Attention for Image Captioning”, IEEE Transactions on Circuits and Systems for Video Technology, 2021. Chenggang Yan, Tong Teng, Yutao Liu, Yongbing Zhang, Haoqian Wang, Xiangyang Ji, “Precise No-Reference Image Quality Evaluation Based on Distortion Identification”, ACM Transactions on Multimedia Computing Communications and Applications 2021.Chenggang Yan, Lixuan Meng, Liang Li, Jiehua Zhang, Jian Yin, Jiyong Zhang, Zhan Wang, Bolun Zheng, “Age-Invariant Face Recognition By Multi-Feature Fusion and Decomposition with Self-Attention”, ACM Transactions on Multimedia Computing Communications and Applications 2021
2. Regarding the writing of the article. Suggest placing previously described, informal, and brief information in the subject position at the beginning of the sentence, and placing new, upcoming, complex, and long information in the emphasized position at the end of the sentence.
1. The manuscript changes the traditional masking method that only relies on one evaluation function, combining multiple evaluation functions to comprehensively evaluate pixel pairs to select high-quality pixel pairs, avoiding the limitation of a single evaluation function.
2. In order to ensure that the matching problem can be solved even with limited computing resources, a new perspective was adopted. The manuscript proposes a new GP MCMatching algorithm, in which we use Gaussian Process Fitting Model (GPFM) instead of objective function to search for the optimal pixel pair. By using this algorithm, effective and accurate matching can be achieved with only 1% of computing resources.
Reviewer 2 Report
The topic of this paper touches on aspects of a new method for an image processing task which separates the foreground from the background in images. The authors try to overcome deficiencies from well-established sampling methods and evolutionary algorithms and decided to approximate the distribution of an objective function to solve a constrained optimization problem more efficiently. After the presentation of previous state-of-the-art work and theoretical basis, they presented the results of their algorithm in one benchmark dataset and presented its superiority when the computational resources are limited. Nonetheless, some cases were documented where it can be shown that the solution has some suboptimal results.
The paper contains enough originality to justify a publication. This is a new method that solves quite important problems and can save lots of resources.
There is enough background work mentioned.
The paper is based on appropriate methodology and theory. Some issues need to be tackled: first, the authors assume that there is an optimal pixel pair present. Although this is a quite valid assumption it would be better to create a synthetic dataset with images where this pair is pre-specified. In lines 153-154 there is a question of how the known vs. unknown regions are defined; does a human label them? Is the range between the pixels (line 183) adaptive? What is going to happen if the assumption in line 191 is not met? The equations (8) – (10) are almost the same; what would happen if they were not or if their sorting (what comes first and what comes second) was swapped? Is the averaging in line 247 the best way of creating the overall evaluation function or can there be a weighted average which is learned more helpful? Who sets the thresholds mentioned on page 8? Is the assumption in lines 275-279 met? The reviewers advise strongly to show that this assumption is met, or at least what happens if it is not met. It is not clear what is the kernel of the Gaussian Process. Is the theoretical assumption in lines 322-323 met? Why Is the optimal pixel pair located in the X^loc of the known region closest to the unknown pixels? Is it always the case? Why and when is the assumption mentioned in line 320 not met? What does the algorithm do then? There are more examples needed for the specification of parameters tau, c and gamma on page 10. The claim in lines 331-332 needs proof, a guarantee and more examples.
It would be quite interesting to show the results with more datasets; the dataset that was used is quite small compared to state-of-the-art image processing datasets. Who sets the value of epsilon in line 361? Are the results depicted in Figure 5 reasonable? Is there an interpretation for them? The reviewers advise the presentation of results and code in a GitHub repository and an evaluation with human experts too. The honesty of suboptimal results is appreciated, but a sketch of how those shortcomings (Conclusion) will be tackled is necessary for acceptance.
The paper is well-written and good organized. No typos or major language problems were found.
Minor editing of English language required
Round 2
Reviewer 2 Report
The reviewer's comments were adequately addressed. Nonetheless, they are not in the revised paper! I highly encourage the authors to add it or at least publish it in an upcoming paper.
Minor editing of English language required